# Semi-connected Joint Entity Recognition and Relation Extraction of Contextual Entities in Family History Records

**Anonymous Institute**

## Abstract

Entity extraction is an important step in document understanding. Higher accuracy entity extraction on fine-grained entities can be achieved by combining the utility of Named Entity Recognition (NER) and Relation Extraction (RE) models. In this paper, a semi-connected joint model is proposed that implements NER and Relation extraction. This joint model utilizes relations between entities to infer context-dependent fine-grain named entities in text corpora. The RE module is prevented from conveying information to the NER module which reduces the error accumulation during training. That improves on the fine-grained NER F1-score of existing state-of-the-art from .4753 to .8563 on our data. This provides the potential for further applications in historical document processing. These applications will enable automated searching of historical documents, such as those used in economics research and family history.

## 1 Introduction

Named Entity Recognition (NER) also called entity extraction or entity identification – is a natural language processing (NLP) technique that automatically identifies named entities (names, places or dates for example) in a text and classifies them into predefined categories. It is often sufficient to identify an entity as a course grained entity like a name if the application is attempting to identify employees working for a company from a paragraph of text. Although this course grained entity recognition is sufficient for many applications, fine grained classification is necessary for family history applications where it is necessary to know that relationship between different entities in addition to deriving their classification.

There are many companies and research organizations in the fields of family history and historical document understanding. Family history work helps people learn about their heritage and form connections with their ancestors. To facilitate their work they often automate the extraction of information en masse from historical documents. One part of this process is called entity extraction. For example, in family history, digital text is searched for particular entities. These entities include names of parents, names of children, birth dates, marriage dates, etc.

These entities are extracted and compared to each other to build family tree charts in a process sometimes called indexing. To precisely index historical documents it is not enough to have course-grained labels, such as name or date, but fine-grained labels, such as spouse name and marriage date, are necessary. Furthermore, these fine-grained labels often rely on the document's internal context.

Unfortunately, these entities are often fine-grained and contextual to relationships between entities within a record. These organizations do not have models that can accurately extract such context-dependent fine-grained entities, so they instead extract important words as coarse-grained entities (such as person, place, or date) and manually label them as fine-grained classifications. This problem is even more pronounced for records written in languages such as French with less labeled data.

## 2 Related Work

In 2018, Belkoulis revolutionized the field of named entity recognition (NER) by conceptualizing joint entity and relationship extraction as a multi-head selection problem (Bekoulis et al., 2018). His paper demonstrated that relation extraction was a helpful tool to improve entity extraction accuracy. His original model used a bidirectional LSTM for sentence encoding. That same year, papers saw improved results using ELMo (Peters et al., 2018) for sentence encoding (Sanh et al., 2019). After the introduction of BERT (Devlin

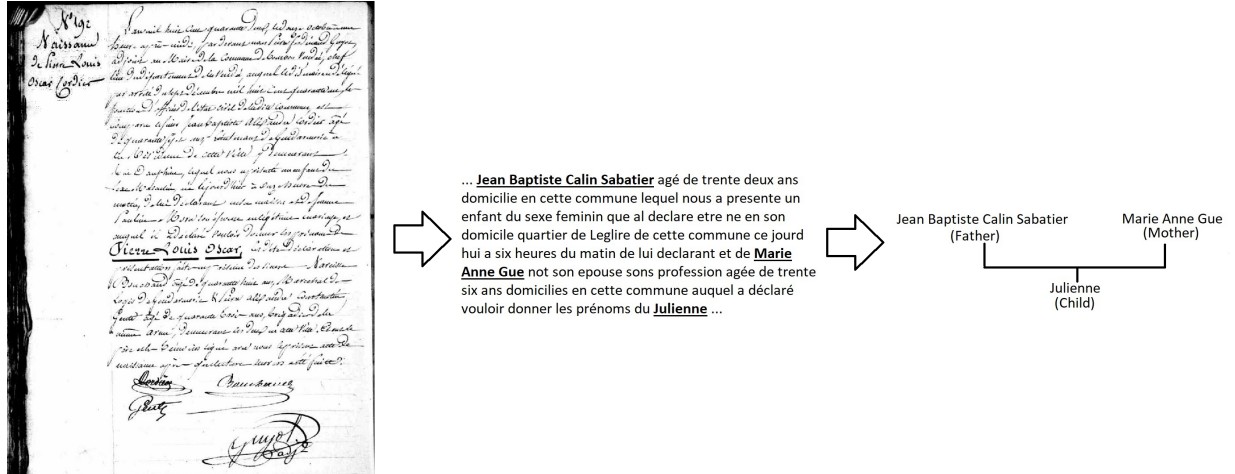

Figure 1: Transcription on records into Family trees. Example of a handwritten birth record, written in French, which after transcription(using the handwriting recognition model) is passed through our joint entity relation model in order to identify the entities and their respective relations. Note that the transcription is not perfect as it contains spelling mistakes, grammatical inconsistencies, etc that make it harder for the model to find correct entities and relations

et al., 2019) in 2019, several papers saw even better results when using BERT for sentence encoding instead of ELMo.

Since then, papers in the field have improved entity recognition accuracy by creating joint models that use relation extraction. The ways these joint models are implemented vary. One approach is to have the joint model ask questions about the data (Li et al., 2019)(Zhao et al., 2020). Another approach uses distantly supervised data augmentation to reduce the impact of negative labels on the joint model (Xie et al., 2021). Jue Wang's paper sees improved relation extraction by filling an entity-relation table (Wang and Lu, 2020). Hierarchical relationship extraction is particularly effective at detecting hierarchical relationships (Han et al., 2018)(Takanobu et al., 2019)(Zhang et al., 2021).

The most successful approaches for entity recognition insert markers into the sentences. This is done in models such as PURE (Zhong and Chen, 2021). These markers reduce the need for embeddings which reduces memory needed and improves inference speed. Papers that use this approach are in the top 3 micro-F1 scores for both entity extraction and relation extraction accuracy for the benchmark datasets: CoNLL 2003, ACE2004, ACE2005, and SciERC (Ye et al., 2021).

The models from existing research perform well on benchmark datasets. However, they fail to perform on more complicated datasets, such as ones

with fine-grained or contextual entities. Most of these models can not find all the relations in a multi-sentence corpus because they can not map a relationship between two entities in separate sentences. However, cross-sentence relation extraction is necessary to find the nuanced relationships in family history records.

## 3 Problem Description

Existing research insufficiently performs the task of fine-grained entity extraction on contextual entities. Traditional methods may be able to identify names and dates, but are unable to identify Mother's Name vs Sisters Name or the dates corresponding to different events in a record. Much of the difficulty comes from the space between entities in a paragraph.

Combining entity recognition with relation extraction significantly improves the accuracy of contextual fine-grained entity extraction for automatic indexing systems used by researchers performing automated historical document analysis. This contributes a novel solution to significantly reduce the manual annotation effort when indexing records without sacrificing recognition accuracy.

The indexing of family history records requires a NER model that identifies contextual fine-grained entities. However, it is difficult to train NER models to do that on their own. Relation extraction can help find the context in the corpus by identifying relations between entities. A joint model is needed

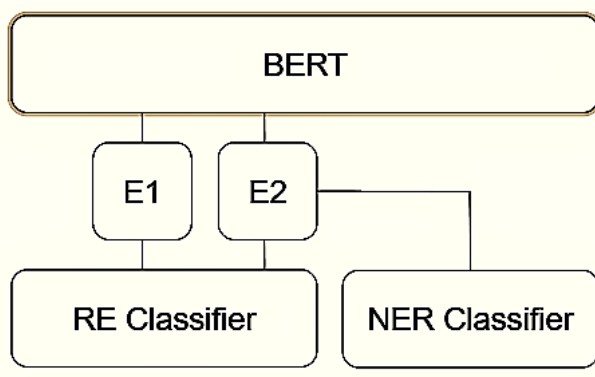

Figure 2: The Joint Entity and Relation Extraction model. The tokens encoded by BERT, Entity 1 (E1) and Entity 2 (E2), are passed simultaneously through the NER and RE module as an end-to-end model to preserve cross-sentence context and prevent the compounding of error.

that can both extract entities and the relationships between those entities. This kind of model more accurately extracts context-based fine-grained entities in family history records.

Additionally, most real-world transcriptions from handwritten documents are full of errors. Common errors include incomplete transcriptions, transcriptions in the wrong language, incorrect labeling, grammar mistakes, and spelling errors. The model built must be able to overcome these difficulties.

## 3.1 Methods

A joint entity-relation extraction model can be used to extract fine grained entities from transcriptions resulting from automated handwriting transcription. A language model is needed to encode tokens and obtain contextualized representations for each input token in the training dataset. The best language model for this is BERT. Most joint entity and relation extraction models suffer from cascading error propagation. In this domain, errors in course grain entity extraction can increase errors in the relation extraction phase of the algorithm. These errors can accumulate more rapidly if the two sides of the joint model are too dependent on each other. To prevent this, the BERT encoded tokens are simultaneously passed through the entity extraction module and the relation extraction module as an end-to-end model (as illustrated in Figure 1). This approach preserves cross-sentence context and reduces the opportunity for errors to compound in the pipeline. Additionally, the NER model and

Relation extraction model both use cross entropy for their loss function in order to improve training accuracy.

For each record, the pipeline labels entities with tokens (as seen in Example 1) and then pairs combinations of tokens as subject and object (as seen in Example 2). It then feeds them to the relation extraction module. For each pair, the module predicts the relation between the two entities and generates a triplet tuple that represents the predicted relation. From those tokens the module classifies the entities and generates tuples that represent the entity classifications. The tuples from the relation extraction module and entity extraction module can be used to infer the fine-grained labels of the entities.

**Example 1:**
{ENTSTART=Name} Paul {ENTEND=Name} and {ENTSTART=Name} James Thompson {ENTEND=Name} ran to the store to get milk for their parents in {ENTSTART=Month} March {ENTEND=Month}.

**Example 2:**
{SUBJSTART=Name} Paul {SUBJEND=Name} and {OBJSTART=Name} James Thompson {OBJEND=Name} ran to the store to get milk for their parents in March.

### 3.1.1 Data Augmentation

It is relatively hard to augment textual data to train a neural model–especially when there is a limited amount of sample data. Given that the data used was French records, many records were fairly similar if not for grammatical differences. These differences lead to the thought and creation of a context-free grammar where rules are mapped to different phrasings found in the sample records. The combination of many different rules, many being recursive rules, and comprehensive dictionaries for simple terms (like names) allows for high variation in creating a new record. Since each rule is created based on real sample data, the generated records are coherent with small seemingly insignificant errors.

Our approach for data augmentation was to create a grammar and templates that implemented the grammar. The grammar consists of both simple and complex rules. Each simple rule is linked to a dictionary of strings and, when parsed, chooses a random element from its dictionary. "SelfGiven-Name", "Surname", and "Profession" are all ex-

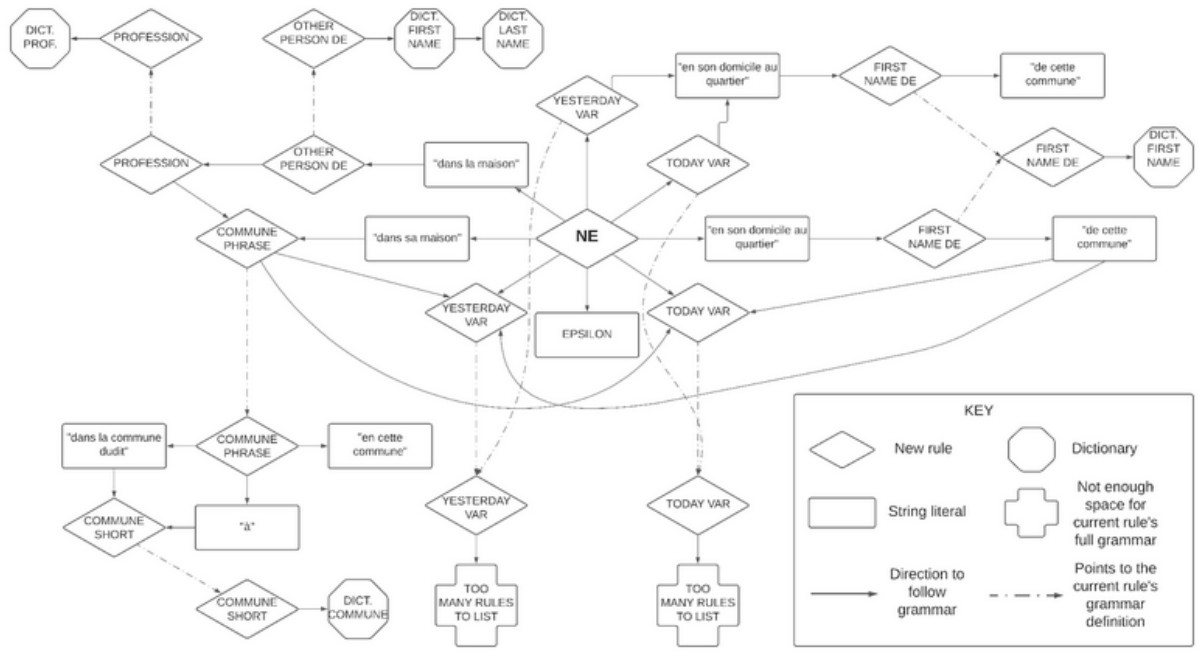

Figure 3: Textual data with similarities can be augmented through the use of a context-free grammar (CFG) and context templates. Creating rule trees (such as in the image above) for the grammar can be done by grouping together different phrases/terms that are encompassed by similar ones. The generated rules, along with the similar phrases/terms left untouched, can be combined to create the context templates. Once templates and rules exist, new textual data can be created. Adding rules increases data variation exponentially.

amples of simple rules. Complex rules, on the other hand, are designed to vary sentence structure. They eliminate the need to create multiple templates which all have a similar record structure. Unlike simple rules, complex rules are not linked solely to a dictionary, but are recursively connected to other rules. Since one rule is linked to many other rules, a diverse amount of data can be generated. An example of one of these complex rules is the "Ne" rule. This rule could appear in the template in a phrase like (English translation given) "...he presented a child of the feminine sex born "NE" of the person speaking and..." The flowchart in Figure 3 illustrates the complexity of this rule.

Diversity within a template can be obtained through the use of these simple and complex rules, but if a record structure is considerably different then a new template needs to be made. There are currently three birth record templates. The first contains the structure of a record which has both a father and mother present, the second template varies in that a midwife is present and is the main speaker in the record, and the third template has no father present but includes the mother's parents.

These templates and rules were generated through the careful consideration of the textual

birth records given to us. Around 300 records were found, but only half of those records were legible. Others contained various errors like non-completion, too many typos, and repetitive, randomly-placed phrases which led to unusable records. The data augmenter should, in theory, be able to generate the content of every legible birth record (excluding typos and small grammatical errors).

In some cases certain phrases need to be cached for later use. For example, if in a marriage record the husband's surname is randomly attributed to "Dupont" then it is natural to expect his father's surname to be "Dupont" as well. A list of cacheable rules must be predefined for this to happen. Each time a rule is compiled, it first needs to check if it is a cache-able rule. If so, it checks if it has already been cached and then uses what is cached. If not, or if the rule was not yet cached, it will compile the rule and cache the result if it should be cached. Each time a new record is started, the cache is cleared.

The generation of a record from a template is a relatively simple process. First, parse the contents of the template to understand exactly what to change and what stays the same. This parsing

process utilizes a predefined set of tags. Each tag has a predefined order of operations that must be performed in order to successfully comply with the language structure and meaning. Each tag is addressed one at a time and a string is eventually generated through its associated rule. Also, rules can be recursive, referencing other rules and tags. Elements of the rules that are string literals stop here, while other rules linked to dictionaries randomly sampled their associated dictionary. Once all the elements of a rule have been augmented/translated, they are bubbled back up to the template where they are injected as a string into the output.

Since the records received were not perfect and had many errors, it is necessary to introduce another process capable of inserting noise into the augmented data in an attempt to mimic the imperfections of the received records. This noise is done character by character and consists of random insertions, deletions, and alterations. Even spacing can be removed between words or inserted into the middle of a word, and yet the correct tag is preserved. The noise can also easily be varied from 0 percent (perfect record) to 100 percent (not a character left unaltered).

### 3.1.2   Named Entity Recognition

Named Entity Recognition is performed using the context-aware embeddings produced by BERT. After running the raw text through BERT, the embedding produced for each word is normalized. This normalization process prevents exploding gradients.

Each embedding is then run through two linear output layers. The first layer predicts BIO tags (Ramshaw and Marcus, 1999). BIO tagging is a common tagging format for tagging tokens in a chunking task in computational linguistics (ex. named-entity recognition). The B- prefix before a tag indicates that the tag is the beginning of a chunk, and an I- prefix before a tag indicates that the tag is inside a chunk. The B- tag is used only when a tag is followed by a tag of the same type without O tokens between them. An O tag indicates that a token belongs to no entity/chunk. The BIO tagging helps keep entities separate even if they are the same entity category. The last layer completes the process and predicts the entity type between Name, Date, Gender, Age, and a None class.

### 3.1.3   Relation Extraction

Initially, our Relationship Extraction model was heavily inspired by the PURE model (Zhong and Chen, 2021). PURE works by adding markers to highlight the two objects within the sentence. These markers help the BERT part of the model extract context-dependent embeddings. As such, these markers reduce the need for embeddings which reduces memory needed and improves inference speed. Then the model classifies relationships based on embeddings for those markers. Training with these markers has the advantage of maintaining the embeddings of other words, which helps the PURE model generalize to unseen words.

However, the historical records differ largely from the records that PURE was trained on. Generally, relationships need to be found between 10-13 entities in those records. Re-encoding the sentence and running it through BERT for each possible pair of entities would cause a large computation overhead. Additionally, these records have a relatively constrained vocabulary, which reduced the generalization benefit of using markers. Due to these differences, the markers are dropped in the modified model.

The relationships are instead identified by concatenating the embeddings of the first word in each entity in a pair and running them through a linear classifier. Dropping the markers also allows for sharing the BERT embeddings between the NER and RE steps, such that the raw text is passed through BERT once. The output of BERT is fed through the two NER outputs mentioned above and the one RE output.

The computation overhead of running this classifier on all possible pairs of words in the sentence is negligible compared to the complexity of BERT itself. Due to this difference, the model can consider all possible relationships for all entity types in an end-to-end fashion. The final confidence of a relationship tuple is the geometric mean of the confidence of the relationship with the confidence of the entities involved.

### 3.1.4   Extracting fine-grained entities

After the relations between entities are found in a record, they can be converted into fine-grained entities. By assigning one person as the primary subject of the record, these entity relationships can be used to convert the coarse-grained labels on entities (such as Name) into fine-grained entities (such as FatherName).

| Entity | Precision | Recall | F1 |
|---|---|---|---|
| Name | 98.92 | 97.40 | |
| Year | 98.62 | 98.62 | |
| Month | 92.41 | 94.81 | |
| Day | 94.12 | 93.02 | |
| Gender | 99.99 | 99.99 | |
| Age | 95.52 | 99.99 | |
| Micro Avg. | 98.49 | 97.25 | 97.87 |

Table 1: Entity recognition accuracy metrics on a set of French birth and marriage dataset from the 19th Century. It is interesting to note that precision and recall are more than 90% for all the entities. The Date entity has been further divided into the year, birth and day and still its accuracy metrics are not much compromised.

For example, if "Ted" is the main subject of the record then:

("Susan", SpouseOf, "Ted") -> (Mary, SpouseName)
("1830", BirthOf, "Ted") -> ("1830", BirthYear)
("Thirty", AgeOf, "Ted") -> ("Thirty", SelfAge)
("Male", GenderOf, "Ted") -> ("Male", SelfGender)

## 4 Experiments

### 4.1 Setup

To verify the effectiveness of joint entity and relation extraction, comprehensive experiments are conducted on a corpus of French birth records and marriage records. Our joint model uses the same training dataset and test dataset for both the NER component and relation extraction component. Using data augmentation on the french records, a dataset of 400 artificial birth and marriage records is generated at the time of execution. This is used as the training dataset. The test dataset is the original 270 hand-labeled french birth records and marriage records. However, this test dataset comes with many complications. These include inaccurate labels, spelling mistakes, missing punctuation, myriads of categories and lengthy sentences. Also French has fewer existing language models and more complicated grammar rules, adding to the complications.

All these anomalies make it harder for neural networks to understand the data. Whereas, the benchmark datasets (in relation extraction) have nearly perfect sentences in English with fewer entity and relation types. These qualities make the benchmark datasets less noisy than our French dataset.

### 4.2 Entity Extraction Results

As seen in Table 1, the rough-grained entity extraction component of our joint model has a total

| Relation | Precision | Recall | F1 |
|---|---|---|---|
| GenderOf | 99.99 | 99.99 | |
| AgeOf | 96.33 | 98.13 | |
| FatherOf | 91.428 | 98.63 | |
| MotherOf | 97.26 | 97.10 | |
| SpouseOf | 92.86 | 99.99 | |
| BirthOf | 99.99 | 99.99 | |
| MarriageOf | 99.99 | 99.99 | |
| Micro Avg. | 96.04 | 98.83 | 97.42 |

Table 2: Relation recognition accuracy metrics on French birth and marriage dataset from the 19th Century. It is interesting to note that precision and recall are more than 90% for all the relations. 'Fatherof' and 'Spouseof' have the lowest precision of all but still, they are at 91.428 and 92.86.

| Model | NER Micro F-1 | Relation Micro F-1 |
|---|---|---|
| **Ours** | **97.9** | **97.4** |
| PL-Marker (Ye et al., 2021) | 91.1 | 73.0 |
| PURE (Zhong and Chen, 2021) | 90.9 | 69.4 |
| Table-Sequence (Wang and Lu, 2020) | 89.5 | 67.6 |
| TriMF (Shen et al., 2021) | 87.6 | 66.5 |
| TablERT (Ma et al., 2022) | 88.0 | 66.1 |

Table 3: Our model performs better than the best models in the field of NER (97.9%) and the field of Relation Extraction (97.4%).

micro F1-score of approximately 97.87%. This is a marked improvement over the benchmark model in Entity Recognition, the PL-Marker model, which has a micro F1-score of 91.1% (Ye et al., 2021). The higher accuracy of our entity recognition model is mutually beneficial to the relation extraction component of our joint model. Of note is our entity extraction architecture is able to successfully train with relatively little training data and perform remarkably well. This also allows our model to train very quickly. One reason the entity extraction is so good, is our use of BERT. Its ability to parse sentences bidirectionally allows for embeddings to be built bidirectionally.

### 4.3 Relation Extraction Results

The benchmark in the field for relation extraction, the PL-Marker model, has a micro F1 score of 73%

| State-of-the-art Fine-grained Entity Extraction (FlairNER) | | | | Our model (Joint NER-Relation Extraction) | | | |
|---|---|---|---|---|---|---|---|
| **Fine-Entity** | **Precision** | **Recall** | **F1** | **Fine-Entity** | **Precision** | **Recall** | **F1** |
| SelfGender | 98.04 | 99.99 | | SelfGender | 99.99 | 96.15 | |
| SelfName | 16.67 | 0.55 | | SelfName | 89.29 | 97.40 | |
| FatherName | 36.84 | 28.00 | | FatherName | 81.25 | 90.28 | |
| MotherName | 42.86 | 45.00 | | MotherName | 89.74 | 92.11 | |
| SpouseName | 41.74 | 47.52 | | SpouseName | 75.86 | 84.62 | |
| SpouseFatherName | 34.12 | 38.16 | | SpouseFatherName | 46.15 | 57.14 | |
| SpouseMotherName | 44.29 | 48.44 | | SpouseMotherName | 56.00 | 58.33 | |
| OtherPersonName | 3.80 | 88.97 | | OtherPersonName | 91.80 | 84.85 | |
| SelfAge | 76.71 | 73.68 | | SelfAge | 75.81 | 99.99 | |
| FatherAge | 51.90 | 49.40 | | FatherAge | 76.92 | 76.92 | |
| MotherAge | 56.07 | 64.52 | | MotherAge | 91.67 | 88.00 | |
| SpouseAge | 83.87 | 72.22 | | SpouseAge | 60.71 | 77.27 | |
| SpouseFatherAge | 33.61 | 16.67 | | SpouseFatherAge | 36.35 | 38.75 | |
| SpouseMotherAge | 35.71 | 35.71 | | SpouseMotherAge | 20.00 | 33.33 | |
| BirthDay | 72.73 | 77.78 | | BirthDay | 66.67 | 99.99 | |
| BirthMonth | 76.09 | 83.33 | | BirthMonth | 66.67 | 99.99 | |
| BirthYear | 90.34 | 81.12 | | BirthYear | 99.99 | 99.99 | |
| MarriageDay | 47.37 | 45.00 | | MarriageDay | 99.99 | 99.99 | |
| MarriageMonth | 53.33 | 30.00 | | MarriageMonth | 57.143 | 99.99 | |
| MarriageYear | 52.46 | 56.14 | | MarriageYear | 50.00 | 99.99 | |
| Micro Avg. | 65.29 | 37.36 | 47.53 | Micro Avg. | **83.98** | **87.35** | **85.63** |

Table 4: Comparison of Fine-grained Entity Extraction between a state-of-the-art NER model (FlairNER) and our proposed joint model. Our model stands as a clear winner with an F1 score being 85.63 whereas the regular NER model's F1 score is 45.53. Note the huge improvement in important fine-grained entities like SelfName FatherName, MotherName, SpouseName, FatherAge, MotherAge. Also, there is a huge improvement in the recall for marriage dates and birth dates.

(Ye et al., 2021). Our model has achieved a RE micro F1 score of approximately 97.42% (see Table 2). This is an exceptional improvement in relation extraction performance. Our model achieves this level of improvement due to the nature of joint entity-relation extraction models where training entity extraction improves relation extraction and to a lesser degree training relation extraction improves entity extraction. Much like the the Entity Extraction component of our model, the Relation Extraction of our model also trains quickly and accurately, with relatively little training data. These impressive results of the relation extraction module (in conjunction with the entity extraction module) clearly demonstrate the value our model has.

### 4.4 Final Results

Next the predicted relationships and entities are used to infer the fine-grained entities. Where as the entity recognition module and relation extraction module avoid compounding errors by using a end-to-end architecture, inferences made after training are very susceptible to compounding errors. This explains why certain fine-grained entities such as the age of the subject's spouse's mother have such low accuracy metrics; they are too sensitive to compounding errors. However, the final results in Table 4 show the joint model is a marked improvement over the default method (vanilla NER) of indexing family history documents.

## 5 Conclusion

The model succeeded in indexing the fine-grained entities in family history records within an acceptable margin of accuracy. Results were validated by recording the model's NER micro F1-score, relation extraction micro F1-score, and the F1 score for each fine-grained entity type at the end of the pipeline. It then compared the NER micro F1-score and RE micro F1-score against the corresponding metrics for benchmark joint entity-relation extraction papers.

The results of our joint model are better than the benchmark models in the field, even though our data was much noisier than the data they used. As such our model contributes to the current literature on joint entity-relation extraction in terms of family history records.

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
