# OpenReview forum: "Semi-connected Joint Entity Recognition and Relation Extraction of Contextual Entities in Family History Records"
_ICLR.cc/2023/Conference — Submitted to ICLR 2023_

### Official Review · Reviewer_AKDM · 2022-10-13

**Confidence:** 4
**Correctness:** 2
**Technical Novelty And Significance:** 1
**Empirical Novelty And Significance:** Not applicable
**Recommendation:** 1

**Clarity, Quality, Novelty And Reproducibility:**

The paper is badly written on structure, motivation and experimental designs. The novelty is so limited. The authors use common methods and basic NLP models, not focusing on specific problem.

**Strength And Weaknesses:**

The motivation is unclear.

1.There is no content about semi-connected. What's the meaning? And why use this structure?

2.The authors said, "Existing research insufficiently performs the task of fine-grained entity extraction on contextual entities." Please explain the contextual entities and fine-grained entities in detail. And give some examples of other models' results.

3.The authors said most of these models could not find all the relations in a multisentence corpus because they can not map a relationship between two entities in separate sentences.  However, there are many cross-sentence relation extraction methods that should be referred.

4.The reason for the usage of BERT is unclear.

5.The authors refer to transcriptions, but there are no related contents, such as tools used and average precision.


The novelty is so limited.

1.The NER and RE jointly learning is a common training strategy, which is not a novel solution. And Why is it more suitable for fine-grained entity extraction?

2.The model structure is simple and common.


The experiments are insufficient.

1.The authors design many rules (around 300), which is helpful for this task. However, it will weaken the learning ability of the model and become an engineering-based method. In addition, the authors should add ablation studies to show the effectiveness of these rules and the BERT basic model when changing other NN structures and removing these rules.

2.For experiments, I seriously doubt that the authors don't run experiments on your datasets using other baseline models because these are easily copied from their paper's results (on the ACE2005 dataset). Therefore, the results in Table 3 are not convincing!



**Summary Of The Paper:**

This paper proposes semi-connected joint entity recognition and relation extraction of contextual entities in family history records ER task. By jointly training NER and RE to improve the fine-grained NER task.

**Summary Of The Review:**

The motivation is unclear. The novelty is so limited. The experiments are insufficient. The experimental results in Table 3 are not comparable and convincing.

---

### Official Review · Reviewer_VUao · 2022-10-28

**Confidence:** 5
**Correctness:** 2
**Technical Novelty And Significance:** 1
**Empirical Novelty And Significance:** 1
**Recommendation:** 3

**Clarity, Quality, Novelty And Reproducibility:**

Clarity:
The claim and results are clear, though not very justified (see weaknesses 3).

Quality:
Overall I found this paper is not well presented (see weaknesses 4).

Novelty:
The novelty is very limited, as the overall model are nearly identically follow existing work, and the proposed augmentation is not only restricted by the "Family History Records" scenario, but also heavily relies on human designed specific rules thus can not generalize.

Reproducibility:
Poor, there is no code available and the only used datasets are custom reserved. Besides, the training protocol including hyperparameters, infrastructure, library is also not explained.

**Strength And Weaknesses:**

Strength:

The paper addresses a new problem of entity recognition from Family History Records.

Comparisons between the proposed method and several related works are made.


Weaknesses:

1. the targeted task, although being newly discussed, is very specialized. I assume it has very limited application scenario and thus this paper provides very restricted reference or insights for most of the community. Besides, the dataset is also not publically available, from the paper I can't get sufficient understanding of this specific task.

2. The proposed method is mostly based on PURE, as clarified in the paper, while the main novelty should be the data augmentation introduced in 3.1.1. This augmentation strategy consists of extremly sophisticated rules that relies on very specific understanding of the structure and content for Family History Records. Overall, I think the proposed method can hardly generalize to very related datasets or even standard benchmark like ACE04/05 etc.

3. The experiments are not justified.

   3.1 The author claims that the proposed method outperforms FlairNER by +38 absolute F1 (Table4), I very much doubt the justification of this conclusion. As two methods are applied with different pipelines. FlairNER directly treats each fine-grained entity as independent categary (I suppose), while the proposed method extract coarse-grained entity and relation, then assemble relation ahead of entity as fine-grained descriptions.

   3.2 For results in Table 3, I also find the comparison hardly convincing, what if "Ours" method are not trained with the augmented data? Such ablation is lacked.

   3.3 Besides, are all those model built upon standard BERT model? Why not use the French version as there already exists one[1]? Will this affects much of the reported results?

4. The presentation need further refine.

   4.1 Introduction section, for many times "coarse" are written into "course".

   4.2 "Hier- archical relationship extraction is particularly ef- fective at detecting hierarchical relationships" This sentence does not make sense to me. And the followed citation should be included in shared parentness.

   4.3 "is considerably different then a new template needs to be made" -> "is considerably different than a new template needs to be made"

[1] Louis Martin, Benjamin Muller, Pedro Javier Ortiz Suárez, Yoann Dupont, Laurent Romary, Éric de la Clergerie, Djamé Seddah, and Benoît Sagot. 2020. CamemBERT: a Tasty French Language Model. In Proceedings of the 58th Annual Meeting of the Association for Computational Linguistics, pages 7203–7219, Online. Association for Computational Linguistics.

**Summary Of The Paper:**

This paper aims to extract "contextual entities" (or fine-grained entities as refered in the paper) from specialized documents, which is Family History Records written in French.

For that, the paper first proposes a heavily cutomized data augmentation method that consists of complex rules and dictionaries.

Then a model for joint extraction of entities and relations is applied, which mostly follows an existing work PURE, only without markers.
The model predicts both entities and relations, based on which fine-grained entities can be identified. Actually, what refered to as fine-grained entites here are entities with adjective relations ahead, such as "FatherAge", "MotherAge", "SpouseAge" etc.

**Summary Of The Review:**

The method is of little novelty; the experiments and comparison are not justifiedly conducted; the task is very specialized and might be of limited interest to the entire community; and the presentation is not well.

Also, the template is not even for ICLR, but ACL instead. I wonder whether this incurs desk rejection.

Therefore, I recommand reject.

---

### Official Review · Reviewer_oJAx · 2022-10-29

**Confidence:** 3
**Clarity, Quality, Novelty And Reproducibility:** IMHO, all four metrics are sub-par.
**Correctness:** 1
**Technical Novelty And Significance:** 2
**Empirical Novelty And Significance:** 2
**Recommendation:** 3

**Details Of Ethics Concerns:**

Family tree data.

**Strength And Weaknesses:**

Strengths:
+ The authors collect a new dataset for French family tree parsing and release it.

Weaknesses:
- The manuscript is not prepared in ICLR template.
- The central technique contribution is not clear to me. It seems that the baseline PURE is also a two-stage method and I fail to understand what 'semi-connected' means.
- The evaluation is not done on public datasets like ACE or SciERC, which makes it difficult to understand where the improvements come from. And I cannot understand why the proposed method can outperform PURE by such a large margin in Table.3. A detailed ablation is recommended.
- If the augmentation strategy based upon context free grammar is the key factor, why not evaluate it on public datasets? Is this feasible?
- I have a layman question: I understand western languages are similar to each other, but is encoding French with a pre-trained BERT really reasonable? Correct me if this is a common practice.

Typos:
- course grain in several places
- Much like the the Entity Extraction component of our model, there are two 'the'.

**Summary Of The Paper:**

*Disclaimer: I am a vision person and an emergency reviewer*

This manuscript studies the problem of named entity recognition (similar to object detection) and relation extraction (similar to human object interaction detection) from French texts transcripted by off-the-shelf OCR models. The method firstly classifies text tokens into pre-defined entity categories then classifies relationships which is quite similar to the two-stage methods we used in human-object interaction detection. A data augmentation method based upon context free parse tree is proposed, but not ablated. Evaluations are done on a newly collected dataset (provided in the supp) and reports improved results on entity F1, relation F1 and combined F1. Codes are not provided or promised.

**Summary Of The Review:**

I fail to understand the core methodology improvement, why evaluating on public datasets is not possible and where the large performance improvement comes from.

---

### Official Review · Reviewer_2SCp · 2022-10-31

**Confidence:** 1
**Clarity, Quality, Novelty And Reproducibility:** No Review
**Correctness:** 1
**Technical Novelty And Significance:** 1
**Empirical Novelty And Significance:** Not applicable
**Recommendation:** 1

**Strength And Weaknesses:**

No Review

**Summary Of The Paper:**

No Review

**Summary Of The Review:**

No Review

---

### Official Review · Reviewer_hDjQ · 2022-12-20

**Confidence:** 1
**Clarity, Quality, Novelty And Reproducibility:** N/A
**Correctness:** 1
**Technical Novelty And Significance:** 1
**Empirical Novelty And Significance:** Not applicable
**Recommendation:** 1

**Strength And Weaknesses:**

N/A

**Summary Of The Paper:**

N/A

**Summary Of The Review:**

This paper uses the ACL template instead of the ICLR one.

---

### Decision · Program_Chairs · 2023-01-20

**Decision:**

Reject

**Justification For Why Not Higher Score:**

There are serious flaws in the paper.

**Justification For Why Not Lower Score:**

N/A

**Metareview: Summary, Strengths And Weaknesses:**

The paper puts forward a new dataset for French family tree parsing.
The application is interesting albeit somewhat narrow. The work, and paper has many limitations along the dimensions of clarity, methodological novelty, motivation of approach, and evaluation. The paper  also fails to follow  ICLR submission guidelines.

**Summary Of Ac-Reviewer Meeting:**

n/a